# Persistently high TB prevalence in Nairobi County neighbourhoods, 2015–2022

Jane R. Ong'ang'o[1]*, Jennifer Ross[2], Richard Kiplimo[3], Cheryl Kerama[1], Khai Hoan Tram[2], Jerry S. Zifodya[4], Nellie Mukiri[5], Erick Nyadimo[1], Martha Njoroge[1], Aiban Ronoh[6], Immaculate Kathure[6], Dickson Kirathe[6], Thomas R. Hawn[2,7], Videlis Nduba[1], David J. Horne[2,7]

1 Centre for Respiratory Diseases Research, Kenya Medical Research Institute, Nairobi, Kenya, 2 Department of Medicine, University of Washington, Seattle, Washington, United States of America, 3 Amref Health Africa, Nairobi, Kenya, 4 Section of Pulmonary, Critical Care and Environmental Medicine, Tulane University, New Orleans, Louisiana, United States of America, 5 National TB Reference Laboratory, Ministry of Health, Nairobi, Kenya, 6 National TB, Leprosy and Lung Disease Programme, Ministry of Health, Nairobi, Kenya, 7 Department of Global Health, University of Washington, Seattle, Washington, United States of America

* jrnabongo@gmail.com

## Abstract

National and sub-national population-based surveys, when performed at intervals, may assess important changes in TB prevalence. In 2022 we re-surveyed nine Nairobi County neighbourhoods that were previously surveyed in 2015. We aimed to determine pulmonary TB prevalence, compare prevalence to 2015 estimates, and evaluate changes in risk groups. Participants who reported cough of any duration and/or whose chest x-ray suggested TB submitted sputum for smear microscopy, Xpert Ultra, and liquid culture. We defined prevalent TB as *Mycobacterium tuberculosis* detection by sputum Xpert or culture, excepting individuals who were only trace positive. Our methods differed from 2015, which used solid media, Xpert MTB/RIF, and cough duration >2 weeks. We calculated TB prevalence using random-effects logistic regression models with missing value imputations and inverse probability weighting. In 2022 among 6369 participants, 1582 submitted ≥1 sputum sample, among whom 42 (2.7%) had TB, a weighted TB prevalence of 806/100,000 (95% confidence interval (CI), 518–1096). An additional 31 (2.0%) participants tested Ultra trace-positive/culture-negative. For comparison to 2015, we excluded 2022 participants (n = 2) whose only criterion for sputum was cough <2 weeks. There was no evidence for a decline in overall TB prevalence from 2015 to 2022. TB prevalence among men was high (1301/100,000) and remained high compared to 2015 (p-value <0.05). The age group with the highest estimated prevalence remained people ages 45–54 years. Among people with prevalent TB who reported cough, 76% had not sought health care. Dissimilar from other serial surveys that showed declines in TB prevalence, we found persistently high TB prevalence over a 7-year period in Nairobi County. Limitations of this study include changes in methodology between the two surveys and complex effects of the COVID-19 pandemic.

**Data availability statement:** Data are available on reasonable request from the custodians who are the Scientific and Ethics Review Unit of Kenya Medical Research Institute (KEMRI) and the National TB, Leprosy and lung disease Programme, all under the Ministry of Health, Kenya. email addresses are as follows; seru@kemri.go.ke and head-nltp@nltp.co.ke.

**Funding:** This research was funded by the National Institutes of Health/NIAID (NIH grant 5R01AI150815 – DJH, TRH, VN), the National Center for Advancing Translational Sciences of the NIH (UL1 TR002319), NIH D43 TW011817-01 (VN, CK, DJH, TRH), and the Firland Foundation (TRH). The funders had no role in study design, data collection and analysis, decision to publish, or preparation of the manuscript.

**Competing interests:** The authors have declared no competing interests exist.

## Introduction

An estimated 10.6 million people fell ill with tuberculosis (TB) globally in 2022, an increase compared to 10.3 million in 2021 [1]. The number of people officially reported with TB in 2022 was 7.5 million, the highest number since global TB monitoring started in 1995. The global gap between estimated and reported number of people with TB, 3.1 million in 2022, contributes disproportionately to TB deaths, TB transmission, and perpetuates challenges in developing TB control policies [2]. There is an urgent need for strategies to increase the proportion of people with TB who are diagnosed and to decrease the time to diagnosis.

Kenya is categorized as one of 30 high TB burden countries globally, with an estimated 128,000 people who developed TB in 2022 [1], among whom more than 40% were not reported to the national TB program. In 2015–16, Kenya conducted a nationwide TB prevalence survey, the first since 1958 [3,4]. Through this effort, there was an upward revision of the national TB prevalence rate to 348 per 100,000 population, compared to a pre-survey estimate of 233/100,000 [3]. Additional findings were that only 46% of Kenyans with TB were diagnosed and started on treatment, with the largest gaps among people 25–34 years and those over 65, and that 65% of people with symptomatic TB had not sought health care prior to the survey.

Population-based TB prevalence surveys offer the most accurate estimates of TB burden [5]. National and sub-national surveys, when performed at short intervals, may assess trends in TB prevalence, evaluate the impact of public health interventions, identify characteristics associated with differences in TB prevalence and inform public health responses. Although several Asian countries [6–9] have recently performed serial national or sub-national TB prevalence surveys using WHO protocols, to our knowledge, this has not occurred in any country on the African continent [10].

Nairobi County, home to Kenya's capital city, reported 10,598 patients diagnosed with drug-susceptible TB in 2021 (13.6% of all Kenyan TB notifications), the highest number of any Kenyan county [11]. We performed a limited prevalence survey by re-visiting nine of the ten geographic clusters in Nairobi County that were previously surveyed in the 2015–16 Kenya national prevalence survey. We aimed to determine the prevalence of bacteriologically-confirmed pulmonary tuberculosis in those 15 years and older, evaluate temporal trends in TB prevalence over a 7-year period (2015 to 2022), and characterize the health care seeking behavior of persons with TB. We hypothesized that TB prevalence would decrease between 2015 and 2022 in the setting of accurate TB estimates from the 2015 survey, increased case notifications since the prior survey, and greater availability of GeneXpert tests for rapid TB diagnosis [1].

## Methods and materials

### Survey design and target population

We performed a cluster-based cross-sectional survey, enrolling participants between 19 May 2022 and 30 November 2022 using the WHO recommended protocol [5]. Each cluster was defined using the same geographical borders as the 2015–16 prevalence survey (Fig 1) [4]. Due to funding, we surveyed nine of 10 clusters included in the 2015–16 survey. Individuals were eligible for enrollment if they were aged 15 years and above, had resided in the surveyed household for at least 30 days prior to the survey date, and provided informed consent.

### Target enrollment

Based on the methods used in the 2015–16 survey [4], a sample size of 6,480 adults was targeted in the nine clusters with an estimated average of 720 adults enrolled from each cluster (allowable cluster range 650–790). All the clusters were urban.

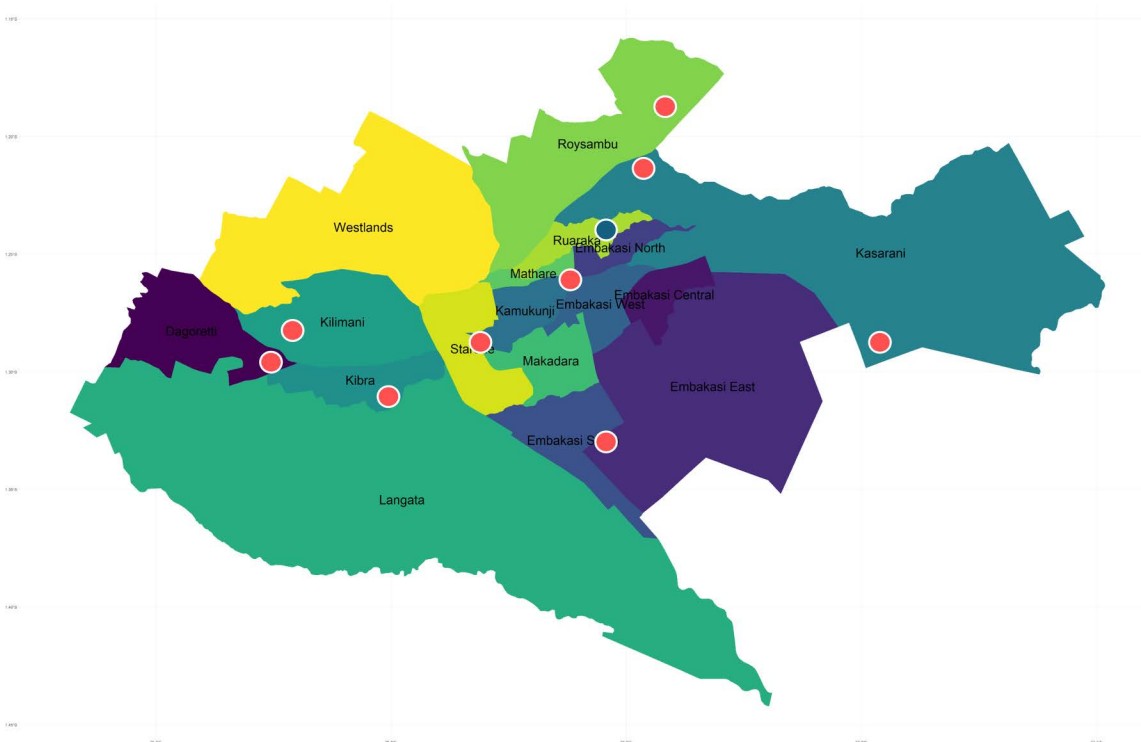

**Fig 1. Map of sub-counties in Nairobi, Kenya, and locations of surveyed geographic clusters (red circles) and excluded cluster (blue circles).** Map was generated in R version 4.2.2 using a shapefile of Kenyan administrative boundaries from geoBoundaries under a CC BY 4.0 license (www.geoboundaries.org).

### Survey field procedures

**Pre-survey procedures.** Prior to deploying the survey field teams, engagement visits were made to appropriate officials at the county and cluster levels to sensitize local authorities on the survey and to obtain their support for survey interventions. This included County Commisioners, Chiefs and Village Elders. Mapping of cluster borders (based on those used in the 2015–16 survey) occurred in this pre-survey phase.

**Survey field procedure at households.** Study clusters were surveyed one at a time using two field-based teams, a mobilization team and an active survey team. The former engaged in community sensitization and mobilization while the latter conducted the survey. In each cluster, the mobilization team performed a door-to-door census using a listing questionnaire to identify eligible residents aged 15 years and above until reaching 650–790 persons. The listing questionnaire collected demographic data on household members regardless of age. A household socioeconomic questionnaire was then administered to each household head. The household-level survey procedures were similar for both surveys [3]. A central team based at the Center for Respiratory Diseases Research at the Kenyan Medical Research Institute (CRDR-KEMRI) provided coordination, monitoring, data management and analysis, and case management.

**Mobile field site procedures.** Eligible participants were invited to the mobile field site where questionnaires, including a symptom screen, and digital chest X-ray (posteroanterior) were performed. Chest X-ray images were independently evaluated on-site by two trained clinical officers using standardized WHO criteria as either "normal", "abnormal suggestive of

TB", or "abnormal other". Study participants who reported a cough of any duration (positive symptom screen) and/or had an abnormal chest X-ray suggestive of TB were eligible for sputum collection. In the 2015–16 survey only participants who reported a cough duration of 2 or more weeks were considered positive on their symptom screen; in the present study we chose a cough of any duration to maximize sensitivity based on the accuracy of cough as a symptom screen for TB [12] while also collecting data on cough duration to allow comparisons between the two surveys. Like the 2015–16 survey, we collected sputum from participants who declined to undergo chest radiographs. Individuals with no radiological abnormalities suggestive of TB and who denied cough did not submit sputum samples.

**Laboratory procedures.** Similar to the 2015–16 survey, we collected two sputum samples from sputum-eligible participants: a "spot" sputum collected at the time of the initial mobile field site visit and a morning sputum, collected by the participant at home on the following morning. These specimens were transported daily under cold chain for processing to CRDR-KEMRI and/or the National Tuberculosis Reference Laboratory (NTRL), both located in Nairobi. Laboratory processes were performed at the CRDR, NTRL, or the CDC/KEMRI laboratory in Kisian, Kenya. We performed direct sputum smears on all samples using Auramine O followed by fluorescence smear microscopy. GeneXpert MTB/RIF Ultra (Xpert Ultra, Cepheid, Sunnyvale, CA, USA) was performed on morning sputum samples or on spot samples when a morning sample was not available. Xpert Ultra results were recorded as cycle threshold (Ct) values for all probes and as a semi-quantitative grade (Negative, Trace, Very Low, Low, Medium, High). The lowest Ct value from any of the four rpoB probes was assigned as the minimum Ct value. For participants with Ultra trace positive results, which are generally positive only for the insertion sequence (IS) probe, we assigned a minimum Ct value of 35 which is near the upper limit of detection. Both spot and morning sputum specimens underwent AFB-culture with Mycobacteria Growth Indicator Tube (MGIT 960, Becton Dickinson, Franklin Lakes, NJ, USA using 7ml tubes supplemented with BD BBL™ MGIT™ OADC and BD BBL™ MGIT™ PANTA, and were incubated in an automated BACTEC MGIT 960 machine for growth determination. Samples that had zero growth units at 42 days were confirmed negative. Broth cultures that were flagged as positive by the MGIT 960 were subjected to brain heart infusion (BHI) agar for sterility and Ziehl-Nielsen staining for the presence of Acid-Fast Bacilli (AFB). Once AFB presence was confirmed, the broth was then subjected to immunochromatographic assay (BD MGIT™ TBc ID, Capillia test) for confirmation of *Mycobacterium tuberculosis* complex (MTBC). For isolates confirmed as MTBC, drug susceptibility testing was done using BD BBL™ MGIT™ SIRE kit and positive isolates were aliquoted for future reference.

Our use of Xpert Ultra differed from the 2015–16 prevalence survey, which used Xpert MTB/RIF, an earlier generation of the assay. To inform comparisons with the earlier survey, we conducted side-by-side testing of Xpert Ultra versus Xpert MTB/RIF in 40 samples selected randomly within semi-quantitative grade strata of Xpert Ultra results. Additionally, culture media differed between surveys as the 2015–16 survey used Lowenstein-Jensen (LJ) solid medium culture while we used MGIT liquid culture in 2022; the latter is a more sensitive culture medium [13].

## Survey TB case definition

We defined a participant as having prevalent pulmonary TB if the Xpert Ultra result was positive at a semi-quantitative grader higher than trace positive and/or at least one sputum culture was positive for *M. tuberculosis*. Different from Kenya Ministry of Health policies, we considered specimens that were Xpert Ultra trace positive and culture negative as negative for TB

(i.e., false positive result). We based this definition on uncertainty around the interpretation of Xpert Ultra trace positive results which may be due to prior TB disease, transient infection, or true positive results among other causes [14–17]. The 2015–16 survey performed PCR-testing using the Xpert MTB/RIF assay, which does not have a trace positive grade, and defined pulmonary TB as any GeneXpert MTB/RIF positive result and/or one or more sputa cultures positive for *M. tuberculosis*. A sub-group of trace-positive participants were enrolled into a study of TB infectiousness [18] and had two additional sputa collected for MGIT culture, one of which underwent Xpert Ultra testing; we report these results. All participants diagnosed with culture-positive TB or with a positive Xpert Ultra test (including trace positive results) were linked to local health facilities for treatment initiation by the survey field coordinator and community health volunteers. The County TB coordinator was informed of all confirmed TB diagnoses and a documented laboratory result form with this information was delivered to the treatment health facility.

## Data collection and management

We collected data in the field using REDCap [19], a web-based application installed on tablet devices. HIV status and diabetes were based on self-report.

## Data analysis

The primary outcome was the prevalence of bacteriologically confirmed pulmonary TB expressed per 100,000 population. Secondary outcome measures were pulmonary TB prevalence by age groups and sex, description of symptoms and radiological findings for pulmonary TB, and the health seeking behaviors of individuals with symptomatic pulmonary TB. Descriptive analyses were done using R software version 4.3.1. All tests were two sided and p-values ≤0.05 were considered significant. The random-effect logistic regression model recommended by the WHO [20] and adopted in the Kenya National TB Prevalence Survey [4] was applied. This model uses robust standard errors with missing value imputations and inverse probability weighting to correct for differentials in participation in the survey by age, sex, and cluster. This model accounts for clustering and variation in the number of individuals per cluster when estimating the point prevalence of pulmonary TB and 95% confidence intervals (CIs) [3]. We used Stata version 15 (StataCorp, College Station, TX) using the mim and svy commands with pweights specified to adjust for design effect. We analyzed data from the 2015 and 2022 prevalence surveys using the same code to generate prevalence estimates. A sensitivity analysis of individuals from the 2022 survey using a cough definition of at least two weeks was performed. To compare case notification rates with prevalence estimates, we calculated case notification rates (CNRs) for each age group from the routine TB data notified to National TB program. These were plotted alongside the prevalence estimates, including the confidence intervals.

## Ethical considerations, consent and confidentiality

This study was approved by the Institutional Review Boards of KEMRI (KEMRI/SERU/3988) and the University of Washington (UW STUDY00009209). After ethical approvals were received, the National Commission for Science, Technology and Innovation (NACOSTI) provided the study with a research clearance permit. Before commencement of the study, administrative approval was sought from the overall Nairobi County administration and Nairobi County health management team.

Written informed consent was obtained from each eligible participant before enrolment in the study and after thorough explanation of the risks and benefits of participating in the

study. Any question raised by the potential participants was answered before participation in the study was offered. It was explicitly stated that participation was completely voluntary, that people could withdraw at any moment, and that there was no obligation to participate. If participants refused to participate or withdraw, they would not lose their rights to whatever benefits they are entitled to in the community. Eligible participants with no or low literacy levels, consent was obtained in the presence of a witness and signature was obtained using their thumb print. For eligible participants below 18 years of age, we obtained guardian consent and participant assent.

To ensure participant confidentiality, a unique participant identification number was assigned to each participant to identify their data. No study participant was identified by name in any report or publication derived from information collected for the study.

### Inclusivity in global research

Additional information regarding the ethical, cultural, and scientific considerations specific to inclusivity in global research is included in the Supporting information (S1 Checklist).

## Results

### Enrollment

From May to November 2022, we conducted a household census in nine Nairobi geographic clusters. We identified 9,459 residents among whom 1,799 were less than 15 years of age (Fig 2). Out of 7,644 persons eligible for enrollment, 83% (n = 6,369) enrolled. This enrollment proportion was lower than the 2015–16 survey, which had 89% (6,834/7,631) enrollment. S1 Table shows the demographic characteristics for people who enrolled and did not enroll in the 2022 survey. The participation rate among men and women was 77% and 89%, respectively. The participation pattern across the age groups was similar for both men and women (S1 Fig).

The median age of study participants in 2022 was 33 years (IQR 24–42) compared to 30 years (IQR 24–39) in 2015.

Age groups skewed to older ages in 2022 compared to 2015 (p-value <0.001) (Table 1). Among enrollees in 2022, the proportion of women (66%) was greater than men (34%). While the proportion of women (59%) was also greater than men (41%) in the 2015–16 survey, women represented a significantly greater proportion in 2022 (p < 0.001). Median household size and the number of participants living alone were 2 (IQR 2–4) and 23.9% in 2022, respectively. There were differences in occupations, marital status and education levels between the two surveys including an increase in reported unemployment from 14% of participants in 2015 to 21% in 2022.

We found that 63% of 2022 survey participants lived in the same location in 2015 (S2 Table). An additional 8% reported living in the same cluster as in 2015–2016 but had moved households. Thirteen percent of participants reported participating in the prior survey. Across clusters, migration was noted to be highest in cluster 2 with 79% of the participants indicating that they did not live in the same household in 2015. The nine surveyed clusters differed in several characteristics including sex distribution and the frequency of participants using tobacco and consuming alcohol (S3 Table).

### Survey screening results

Based on a cough of any duration and/or abnormal chest X-ray consistent with TB, 2,043 participants were screen positive (32.1% of 6369 persons screened) and eligible to submit sputum samples in the 2022 survey. Among these participants, 1,018 (49.8%) were eligible by chest

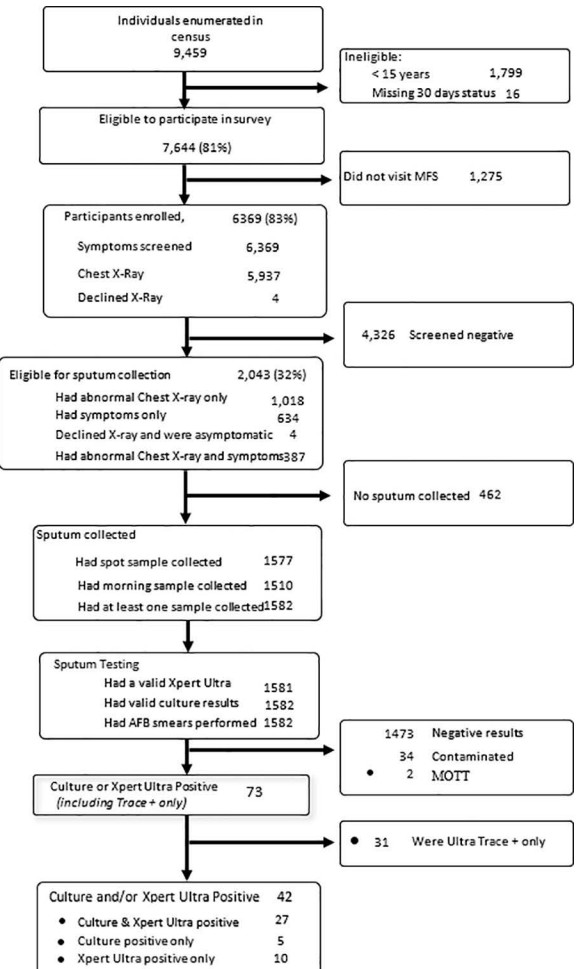

**Fig 2. Schematic diagram of participant enrollment and screening in the 2022 prevalence survey.**

X-ray alone, 634 (31.0%) eligible based on the presence of a cough, and 387 (18.9%) eligible by both screening methods. An additional 4 study participants who did not receive a chest X-ray and did not have cough symptoms were eligible for sputum collection (Fig 2). Cough of any duration was reported by 1,021/6,369 (16%) participants and was the most frequently reported symptom, followed by chest pain (9.8%), and cough >2 weeks (5.7%) (Table 2). Cough of any duration, fever, and chest pain were less frequent in the 2022 prevalence survey compared to 2015. The presence of cough of any duration was similar (p-value 0.07) for men (15%) and women (17%). Cough frequency was higher (p < 0.001) among older age groups (22% for 45–54 years, 23% for 55–64 years, 25% for ≥65 years) than younger age groups (14% for 15–24 years, 13% 25–34 years, 16% for 35–44 years).

## TB prevalence in 2022

Of the 2,043 participants eligible to submit sputum, 1,582 (77.4%) submitted at least one sputum sample. There were 42 (2.7%) persons who met the study definition of TB, among whom 27 were culture and Xpert Ultra positive, 5 were culture-positive only, and 10 were Xpert Ultra positive only (Fig 2). Twenty-seven of the participants with pulmonary TB had smear-positive

**Table 1. Social and demographic characteristics of Nairobi prevalence survey participants in 2015 and 2022.**

| Characteristic | 2015, N = 6,132[1] | 2022, N = 6,369[1] | p-value[2] |
|---|---|---|---|
| Age group | | | <0.001 |
| 15–24 | 1,694 (28%) | 1,662 (26%) | |
| 25–34 | 2,134 (35%) | 1,902 (30%) | |
| 35–44 | 1,199 (20%) | 1,450 (23%) | |
| 45–54 | 642 (10%) | 751 (12%) | |
| 55–64 | 287 (4.7%) | 392 (6%) | |
| 65+ | 176 (2.9%) | 212 (3%) | |
| Age | 30 (24–39) | 33 (24–42) | |
| Sex | | | <0.001 |
| Female | 3,622 (59%) | 4,182 (66%) | |
| Male | 2,510 (41%) | 2,187 (34%) | |
| Diabetes | | 328 (5%) | |
| PLHIV | | | 0.10 |
| Positive | 186 (3%) | 137 (2.2%) | |
| Negative | 4,295 (63%) | 4,014 (63%) | |
| Did not disclose | 2,353 (34%) | 2,218 (35%) | |
| Number of household members | | 2 (2–4) | |
| Participant lives alone (%) | | 1521 (24%) | |
| Occupation | | | <0.001 |
| Self-employed | 2,903 (47%) | 2774 (44%) | |
| Employed by government | 170 (2.8%) | 84 (1%) | |
| Employed in private sector | 886 (14%) | 613 (10%) | |
| Pupil/student | 543 (8.9%) | 586 (9%) | |
| Housewife | 646 (11%) | 880 (14%) | |
| Unemployed | 873 (14%) | 1,309 (21%) | |
| Other | 111 (1.8%) | 120 (2%) | |
| Unknown | 0 | 3 | |
| Marital status | | | <0.001 |
| Single (never been married) | 1,868 (30%) | 2,134 (34%) | |
| Married | 3,796 (62%) | 3,817 (60%) | |
| Divorced/Separated | 307 (5.0%) | 262 (4%) | |
| Widowed | 161 (2.6%) | 155 (2%) | |
| Missing | 0 | 1 | |
| Level of education | | | <0.001 |
| No Schooling | 145 (2.4%) | 166 (3%) | |
| Primary School, Not Completed | 1,041 (17%) | 723 (11%) | |
| Completed Primary School | 1,480 (24%) | 1,223 (19%) | |
| Secondary School, Not Completed | 937 (15%) | 1,266 (20%) | |
| Completed Secondary School | 1,635 (27%) | 2,280 (36%) | |
| Further education after Secondary School | 894 (15%) | 710 (11%) | |
| Missing | 0 | 1 | |

[1]Median (IQR) or Frequency (%).

[2]Pearson's Chi-squared test.

**Table 2. Comparison of symptoms characteristics of the study participants between 2015 and 2022 surveys.**

| Characteristic | 2015, N = 6132[1] | 2022, N = 6,369[1] | p-value[2] |
|---|---|---|---|
| Cough of any duration | 1,018 (17%) | 1,021 (16%) | 0.005 |
| Cough (>2weeks) | 292 (4.8%) | 366 (6%) | 0.20 |
| Missing | 0 | 1 | |
| Chest pain | 782 (13%) | 625 (10%) | <0.001 |
| Fever | 112 (1.8%) | 99 (2%) | 0.03 |
| Night sweats | 160 (2.6%) | 140 (2%) | 0.09 |
| Weight loss | 31 (0.5%) | 93 (2%) | <0.001 |
| Shortness of breath | 406 (6.6%) | 239 (4%) | <0.001 |
| Number of symptoms present | | | <0.001 |
| No symptoms | 4,501 (73%) | 4,958 (78%) | |
| Only 1 symptom | 979 (16%) | 839 (13%) | |
| Only 2 symptoms | 403 (6.6%) | 330 (5%) | |
| 3 or more symptoms | 249 (4.1%) | 242 (4%) | |

[1]Median (IQR) or Frequency (%).

[2]Pearson's Chi-squared test.

sputum. There were 31 participants who were Xpert Ultra trace positive and culture-negative, representing 2.0% of all participants who submitted sputum; these persons did not meet the study definition of pulmonary TB. There was one participant who was taking TB treatment at the time of the survey. For comparison to 2015, we conducted an analysis of the 2022 data based on "restricted eligibility" that excluded participants whose only eligibility criterion to submit sputum was cough <2 weeks, resulting in a decrease of 3 persons with TB (n = 39).

In the nine clusters in 2022, the crude TB prevalence was 659/100,000 (95% CI, 487/100,000–892/100,000) and the crude prevalence of smear-positive pulmonary TB was 294/100,000 (95% CI, 185/100,000–466/100,000). Crude prevalence point estimates were higher in seven of the nine clusters in 2022 relative to 2015 (S4 Table), although these differences were not significant. When the symptom screen was limited to cough ≥2 weeks, 2022 crude TB prevalence was 612/100,000 (95% CI, 447/100,000–838/100,000) which, compared to the 2015 crude prevalence of 538/100,000 (95% CI, 383/100,000–756/100,000), was a non-significant increase in the point estimate of 14%.

In 2022, the weighted prevalence of TB was 806/100,000 (95% CI, 518/100,000–1096/100,000) (Table 3). There were 27 and 37 participants with smear-positive and Xpert Ultra positive TB in the 2022 survey, respectively. The estimated weighted prevalence of smear-positive TB was 583/100,000 (95% CI, 301–866/100,000) and of Xpert Ultra positive TB was 602/100,000 (95% CI, 350–854/100,000). The weighted prevalence of all TB was higher (p-value = 0.03) among men (1394/100,000, 95% CI 802/100,000–1985/100,000) than among women (455/100,000, 95% CI 179/100,000–730/100,000). Similar to findings from 2015 in the nine clusters, the age group weighted TB prevalence was highest for persons 45–54 years of age (Table 3) who had had the largest difference between estimated prevalence and case notification rates (Fig 3).

The weighted TB prevalence in 2022 based on restricted eligibility was 761/100,000 (95% CI 450/100,000–1,073/100,000), compared to 567/100,000 (95% CI, 267/100,000–866/100,000) in 2015 (p = 0.41). In stratified analyses, the point estimates of TB prevalence among men and among women were each 43% higher in 2022 than in 2015, the difference in men was statiscally significant while that of women was not (Table 3 and Fig 4). Similarly, the point estimates of TB prevalence in each age group were higher in 2022 than in 2015.

**Table 3. Estimated weighted pulmonary TB prevalence in 2022 survey by cough duration and in 2015 survey.**

| Robust standard errors with multiple imputation and inverse probability weighting | 2022: cough any duration | 2015: cough >2 weeks | 2022: cough >2 weeks | P-value comparing 2022 restricted analysis to 2015 |
|---|---|---|---|---|
| | Bacteriologically confirmed (N = 42) Point estimate (95%CI) | Bacteriologically confirmed (N = 33) Point estimate (95%CI) | Bacteriological confirmed (N = 39) Point estimate (95%CI) | |
| Overall 1 | 806 (518, 1,096) | 567 (267, 866) | 761 (450, 1,073) | 0.41 |
| Smear-positive TB 3 | 583 (301, 866) | 316 (116, 517) | 557 (303, 811) | 0.17 |
| GeneXpert positive TB 3 | 602 (350, 854) | 491 (234, 747) | 556 (309, 804) | 0.70 |
| Sex 2 | | | | |
| Women | 455 (179, 730) | 306 (107, 505) | 439 (215, 664) | 0.34 |
| Men | 1394 (802, 1,985) | 908 (388, 1,429) | 1301 (734, 1,868) | <0.001 |
| Age 3 | | | | |
| 15–24 years | 407 (117, 697) | 339 (142, 436) | 449 (126, 773) | 0.43 |
| 25–34 years | 853 (251, 1,456) | 597 (174, 1,021) | 791 (311, 1,272) | 0.18 |
| 35–44 years | 875 (335, 1,414) | 615 (229, 1,003) | 814 (204, 1,425) | <0.001 |
| 45–54 years | 1,375 (361, 2,390) | 880 (75, 1,685) | 1,164 (452, 1,875) | <0.001 |
| 55–64 years | 1,115 (209, 2,021) | 832 (127, 1,537) | 1,102 (129, 2,074) | 0.44 |
| 65 years and older | 479 (0, 1,433) | 437 (0, 1,443) | 579 (0, 1,609) | 0.27 |

Variables included in this model: Cough of any duration, night sweats, age group, sex, chest X-ray field reading, weight loss.

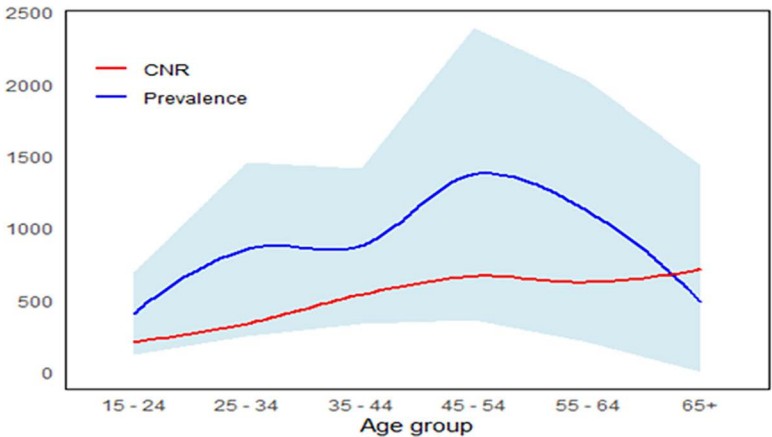

**Fig 3. Comparison of Nairobi TB case notification rate 2022 and estimated TB prevalence in 2022 by age group.**

## Participants with TB: Characteristics, symptoms, CXR

Forty-two participants met the survey definition of TB, including 27 (63%) men, 4 (11%) persons living with HIV, and 1 (2.3%) with diabetes. In addition, 5 (12%) participants reported current smoking while 13 (29.4%) reported alcohol use (Table 4). Among participants with prevalent TB, 64% (n = 27) denied cough of two or more weeks and 40% (n = 17) denied cough of any duration (Table 4). Among participants with TB and cough, the median cough duration was 2 weeks (IQR: 1–3). The WHO recommends a four symptom screen for pulmonary tuberculosis that is based on the presence of one or more of cough, fever, night sweats, or weight loss. Using the WHO screen, 62% (n = 26) of participants with TB would have screened positive.

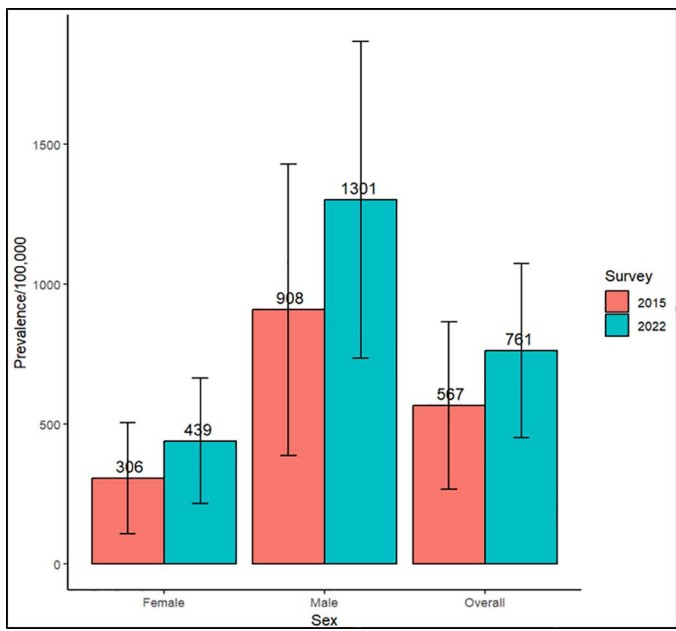

**Fig 4. Weighted TB prevalence by sex in 2015 and 2022 surveys.**

## Participants with Xpert ultra trace positive results

Notably, 31 participants had sputum that was Xpert Ultra trace positive and culture-negative; this represents 2.0% of participants who submitted sputum. There were no samples that were trace positive and culture-positive based on the prevalence survey protocol (Table 4). Among Xpert Ultra positive/culture-negative participants, cough of any duration was the most common symptom and present in 21/31 (62%), which was similar in frequency to participants who were diagnosed with prevalent TB, among whom 25/42 (58%) reported cough (Table 4). Aside from coughing, the second most common symptom reported by the 2 groups was chest pain. A prior history of TB was reported by 7% (2/31) of participants with trace positive/culture-negative sputum. Seventeen of the 31 trace-positive/culture-negative participants were referred to a study in which they had two additional sputa collected for liquid-media based culture. Of these 17 participants, all re-tested as Xpert Ultra trace positive and four were culture positive on one of the additional sputum samples (24%).

## Laboratory/microbiologic results

Rifampicin resistance was not detected by Xpert Ultra or phenotypic drug susceptibility testing (DST). DST identified 2 participants with isolates resistant to streptomycin and isoniazid and one isolate resistant to ethambutol. Among the 68 participants with Xpert Ultra results that were trace positive or higher, 31 (46%) were trace positive, 8 (12%) very low, 14 (21%) low, 6 (9%) medium, and 9 (13%) high semi-quantitative grade. An increasing percentage of samples were culture positive with increasing Xpert Ultra grade; those with "very low" positivity had 25% culture positivity as compared to 100% culture positivity in those with "high" positivity on Xpert Ultra (S5 Table).

## Xpert ultra compared to GeneXpert MTB/RIF

In comparative analyses of Xpert Ultra and GeneXpert MTB/RIF, 40 samples were selected. Of the 13 samples that were trace positive by Xpert Ultra testing, only 1 was positive on

**Table 4. Characteristics of participants with prevalent TB or Xpert ultra trace results.**

| Characteristic | Culture-positive, Ultra negative N = 5[1] | Culture-positive, Ultra positive N = 37[1] | Culture-negative, Ultra trace positive, N = 31[1] | Overall, N = 73[1] | p-value |
|---|---|---|---|---|---|
| Sex | | | | | 0v.3 |
| Male | 2 (40%) | 25 (68%) | 17 (55%) | 44 (60%) | |
| Female | 3 (60%) | 12 (32%) | 14 (45%) | 29 (40%) | |
| Age group | | | | | >0.9 |
| 15–24 | 1 (20%) | 4 (11%) | 5 (16%) | 10 (14%) | |
| 25–34 | 2 (40%) | 11 (30%) | 9 (29%) | 22 (30%) | |
| 35–44 | 2 (40%) | 9 (24%) | 8 (26%) | 19 (26%) | |
| 45–54 | 0 (0%) | 8 (22%) | 5 (16%) | 13 (18%) | |
| 55–64 | 0 (0%) | 4 (11%) | 3 (9.7%) | 7 (9.6%) | |
| 65+ | 0 (0%) | 1 (2.7%) | 1 (3.2%) | 2 (2.7%) | |
| HIV status | | | | | 0.9 |
| Negative | 4 (100%) | 17 (85%) | 15 (79%) | 36 (84%) | |
| Positive | 0 (0%) | 3 (15%) | 4 (21%) | 7 (16%) | |
| Unknown | 1 | 17 | 12 | 30 | |
| Diabetes | 4 (80%) | 36 (97%) | 29 (94%) | 69 (95%) | 0.2 |
| Prior Hx of TB | 0 | 1 (3%) | 2 (7%) | 3 (4%) | >0.9 |
| Smoking | | | | | 0.3 |
| Never | 5 (100%) | 32 (86%) | 28 (90%) | 65 (89%) | |
| Current smoker | 0 (0%) | 5 (14%) | 1 (3.2%) | 6 (8.2%) | |
| Former smoker | 0 (0%) | 0 (0%) | 2 (6.5%) | 2 (2.7%) | |
| Alcohol | | | | | 0.02 |
| Never | 3 (60%) | 26 (70%) | 31 (100%) | 60 (82%) | |
| Once a month or less | 0 (0%) | 2 (5.4%) | 0 (0%) | 2 (2.7%) | |
| 2 to 4 times a month | 1 (20%) | 2 (5.4%) | 0 (0%) | 3 (4.1%) | |
| 2 to 3 times a week | 1 (20%) | 5 (14%) | 0 (0%) | 6 (8.2%) | |
| 4 or more times a week | 0 (0%) | 2 (5.4%) | 0 (0%) | 2 (2.7%) | |
| Reported symptoms | | | | | |
| Cough of any duration | 4 (80%) | 21 (57%) | 21 (68%) | 46 (63%) | 0.6 |
| Cough > 2wks | 1 (20%) | 14 (39%) | 8 (29%) | 23 (33%) | 0.6 |
| Unknown | 0 | 1 | 3 | 4 | |
| Fever | 0 (0%) | 4 (11%) | 2 (6.5%) | 6 (8.2%) | 0.8 |
| Weight loss | 1 (20%) | 4 (11%) | 1 (3.2%) | 6 (8.2%) | 0.2 |
| Night sweats | 1 (20%) | 6 (16%) | 2 (6.5%) | 9 (12%) | 0.3 |
| Fatigue | 2 (40%) | 8 (22%) | 5 (16%) | 15 (21%) | 0.4 |
| Shortness of breath | 0 (0%) | 6 (16%) | 7 (23%) | 13 (18%) | 0.6 |
| Chest pain | 3 (60%) | 12 (32%) | 13 (42%) | 28 (38%) | 0.4 |
| One or more of WHO screening symptoms (cough/fever/weight loss/ night sweats) | 4 (80%) | 22 (59%) | 21 (68%) | 47 (64%) | 0.6 |
| X-ray findings | | | | | 0.007 |
| Normal | 2 (40%) | 2 (5.6%) | 10 (33%) | 14 (20%) | |
| Abnormal, suggestive of TB | 3 (60%) | 34 (94%) | 20 (67%) | 57 (80%) | |
| Unknown | 0 | 1 | 1 | 2 | |

[1]Median (IQR) or Frequency (%).

GeneXpert MTB/RIF (S6 Table). Fourteen (82%) of the 17 Xpert Ultra participants with positive results> trace were also test positive by GeneXpert MTB/RIF; thus, Xpert Ultra detected 3 additional persons with tuberculosis. Two of these participants had semi-quantitative grades of very low on Xpert Ultra, and one was low. All the negative Xpert Ultra participants were also negative on GeneXpert MTB/RIF with valid results. On further comparison of Xpert Ultra and GeneXpert MTB/RIF classification of positivity by cycle threshold, GeneXpert classes correlated well.

## Discussion

In a population-based study of pulmonary TB prevalence in nine geographic clusters in Nairobi County, Kenya, we found that the estimated prevalence of TB remained high from 2015 to 2022. Despite programmatic interventions after the 2015 survey, there was no evidence of a decline in TB overall, by sex, or in age subgroups, a finding that differs from other serial surveys in high TB burden settings during similar time periods. We also found that men and 45–54 year old participants belonged to groups with higher risk for TB in these geographic clusters in both 2015 and 2022. This suggests that in Nairobi County, local drivers of the TB epidemic are stable and may be used to develop targeted TB control interventions. Without specific interventions, it is likely that TB burden will remain high.

After 2015 TB prevalence survey results revealed that 40% of people with prevalent TB had not been previously diagnosed [3], Kenya initiated public health interventions that are ongoing to close the gap in TB case detection. This included greater access at government clinics to GeneXpert and other WHO recommomended rapid diagnostic tests as first-line TB diagnostics, increased use of digital X-ray for screening with CAD interpretation, strengthened processes for community referrals of persons presumed to have TB to government clinics, more efficient contact investigations, and programmatic support for quality improvement at the health facility level. While there were challenges in implementing these initiatives, such as erratic availability of Xpert cartridges, the estimated frequency of undetected TB reduced to 32% in 2022 [21]. Concerningly, we found that the prevalence of TB in re-sampled Nairobi clusters remained unchanged from 2015 to 2022, with higher point estimates overall and in seven of nine sampled clusters; in addition, only one participant with TB was on treatment at the time of our survey.

In comparing our results from 2022 and 2015, it is important to note that the TB detection methods used in 2022 were more sensitive than those used in 2015. Liquid broth-based culture is estimated to be approximately 16% more sensitive than culture on solid media [13], and a meta-analysis of studies that directly compared GeneXpert assays found that, compared to culture, the sensitivity of Xpert Ultra and Xpert MTB/RIF were 90.9% and 84.7%, respectively, a difference of 6.3% [16]. In a limited head-to-head comparison, we also found that Xpert Ultra detected more people with TB than Xpert MTB/RIF. Beyond differences in testing, additional factors likely contributed to the persistently high TB burden in 2022. There were health services disruptions in the two years immediately preceding the 2022 survey, including a COVID-19 related decrease in TB case notifications and a Kenyan healthcare worker strike in December 2020 – January 2021 [22]. The number of notified tuberculosis cases was one of many Kenyan key health indicators that worsened with the onset of the pandemic, decreasing by 26.6% (14.7–45.1%) from March to April 2020. While TB case notifications subsequently increased [1], disruptions in TB diagnosis and treatment associated with these events likely contributed to the persistently high TB prevalence observed in 2022. Similar COVID-19 related disruptions on public health services and their impact on TB control were observed globally in diverse settings including TB endemic and non-endemic nations. We also observed

a significant increase in participants who were unemployed in 2022 compared to 2015, which may predispose to economic deprivation and higher TB risk. In other settings, economic growth has been associated with declines in TB prevalence between sequential surveys [23]. This change in employment may have been due to COVID-19-related effects.

The lack of decline in TB prevalence over a seven-year span differs from other studies conducted in high burden settings where prevalence rates decreased over time [7,24,25]. In Kampala, Uganda, two community active case-finding campaigns that were conducted 2 years apart, the first of which occurred in 2019 prior to the pandemic, found that TB prevalence decreased 45%. While the Ugandan surveys straddled the pandemic, similar to our study, there were several key differences from our study including shorter survey intervals, sputum collection from all participants regardless of symptoms, and a community mobilization component. In addition, individual participation in both the Ugandan surveys was higher (33%–37%) than in our Nairobi survey (13%). These differences raise questions around the most effective time intervals for active case-finding, the durability of reductions in TB prevalence, and the importance of identifying persons with sub-clinical TB and unremarkable chest x-rays.

There remains a high burden of undiagnosed TB in Nairobi and cost-effective active case-finding strategies are needed. Among survey participants diagnosed with bacteriologically confirmed TB, 40% did not report a cough and 38% denied all of the WHO screening symptoms. In our study, chest x-ray demonstrated higher sensitivity as 88% of participants with confirmed TB had an abnormal chest X-ray suggestive of TB. This finding, in agreement with results from other prevalence surveys [3,26], emphasizes importance of chest imaging for detecting persons with prevalent TB in the community [27,28]. However, it is important to note that we likely missed participants with TB who were without cough or abnormal chest x-ray as we would not have collected sputum from these participants. There is growing recognition of the importance of sub-clinical or asymptomatic TB as a large proportion of prevalent TB and as a likely driver of ongoing TB transmission [29,30]. Achieving early diagnosis and treatment in persons with TB without symptoms (with or without radiographic abnormalities) will require novel methods to address diagnostic gaps including point-of-care testing, high population diagnostic yield, and relevance to active case-finding [2].

Development of cost-effective TB case-finding strategies will need to account for behavioral patterns and local variations in epidemiology. Through door-to-door enrollment, our 2022 TB prevalence survey enrolled twice as many women as men, similar frequencies to other surveys [10], yet weighted TB prevalence was three times higher among men than women, again similar to other surveys [24,31]. This suggests that active case-finding approaches will need to identify strategies to increase inclusion of men such as enrollment through non-household locales or performing screening outside of business hours. Further studies of activity spaces and TB risk may help to inform this work. By age group, we verified 2015 findings that older age groups, especially 45–54 years, in Nairobi clusters have the highest TB prevalence which differs from aggregated country-wide data in Kenya (highest prevalence among 25–34 year-olds), and national data from 11 other African nations [10]. Stability of this risk factor among Nairobi clusters supports the development of interventions specific to Nairobi clusters. Intentionally performing active case-finding among 45–64 year-old men in Nairobi County may be a strategy to maximize the detection of TB.

Based on our study definition of TB, sensitivity of Xpert Ultra performed on a single sputum sample (88%) was no different than that of MGIT culture performed on two specimens (76%). Relying solely on Xpert Ultra in our study would have missed 5 of the 42 persons (12%) with confirmed with TB. The sensitivity of a single Xpert Ultra test to that of a single MGIT culture is similar [32], and it has been suggested that primary use of Xpert Ultra with culture for

confirmatory testing in prevalence surveys may be a cost-effective approach [17]. We observed that Ultra trace-positive/culture-negative results were identified in 2% of participants who submitted sputum. Other community-based surveys found that of participants who submitted sputum, 0.4% to 2% had Xpert Ultra trace positive results and that 14% to 43% of the trace positive samples were positive using one to two specimens for culture [14,32,33]. Different from these findings, under our study protocol we did not detect any culture positive samples from participants with Xpert Ultra trace-positive results (n = 31). Seventeen of these 31 participants enrolled in a study of TB infectiousness [18] and provided spot and morning sputum samples: all were trace positive on Xpert Ultra retesting and four (24%) were culture positive on one of two specimens. Based on these findings we suggest that trace positive results be investigated with high quality sputum collection that is submitted for culture, and that multiple sputa submitted for culture is important in confirming culture status. Most persons with trace positive Xpert Ultra sputum who are tested through active case-finding will be culture negative with an unknown role for TB treatment.

There are a number of limitations in our study. While the 2015 survey was powered to sample clusters across Kenya in order to provide an estimate of country-wide prevalence, our 2022 survey targeted only clusters located within Nairobi County. The study focus on Nairobi County does not allow for comparisons with other regions of the country. However, in addition to the findings from the prevalence surveys, there is indirect evidence of regional differences in Kenya. The annual risk of TB infection (ARTI) in Kenya was last described in 2004, at which time Nairobi District had the highest point estimate suggesting lesser impacts of TB control efforts on transmission compared to other districts at that time [34]. Our findings pertain to geographic clusters that are relatively stable in terms of residency as 63% of participants reported living in the same cluster in 2015. Our study enrolled participants in the wake of the COVID pandemic and it is unclear how this impacted TB prevalence in the surveyed clusters. Future work should explore factors such as overcrowding, poor ventilation, poverty, and limited access to healthcare in these neighborhoods to help identify more targeted interventions to address the root causes driving TB transmission in these high-risk areas.

The Nairobi 2022 prevalence survey presents a major strength by returning to previously sampled areas to examine the evolution of TB prevalence over time. As a repeat survey of TB prevalence it allows for assessing trends in TB prevalence within a time interval of at least five years for Nairobi County. Comparing TB prevalence across time allows public health authorities to examine the trend in burden, evaluate the impact of TB control activities between surveys, and create policies to direct future actions. Another unique aspect of this survey was that it was conducted in the backdrop of the COVID-19 pandemic and aimed to provide useful insights on TB care in the setting of a pandemic. This information would be crucial for programmatic management.

## Supporting information

**S1 Fig. Eligible survey population versus Nairobi County Population by age and sex.**
(TIF)

**S1 Table. Demographic characteristics of people who enrolled or did not enroll in the 2022 prevalence survey.** [1]Frequency (%).
(XLSX)

**S2 Table. Participation in the 2015 prevalence survey among 2022 survey participants.**
(XLSX)

**S3 Table. Social and demographic characteristics by cluster.**
(XLSX)

**S4 Table. Participants diagnosed with TB in 2022 and 2015 by geographic cluster.**
(XLSX)

**S5 Table. Xpert Ultra and Culture results among the diagnosed TB cases (including Trace results).**
(XLSX)

**S6 Table. Comparison of GeneXpert results from Xpert Ultra cartridge and Xpert MTB/ RIF cartridge.**
(XLSX)

**S1 Checklist. Ethical considerations and inclusivity in research.**
(DOCX)

## Acknowledgments

We give thanks to individual study participants and their families. We acknowledge the CRDR (KEMRI) study staff that supported data collection, the Kenya National TB Program and the Nairobi county health management team for facilitating the conduction of this study. We also thank all community resource persons who supported the community engagement activities of the study.

## Author contributions

**Conceptualization:** Jane R. Ong'ang'o, Jennifer Ross, Richard Kiplimo, Cheryl Kerama, Khai Hoan Tram, Jerry S. Zifodya, Aiban Ronoh, Immaculate Kathure, Thomas R. Hawn, Videlis Nduba, David J. Horne.

**Data curation:** Jennifer Ross, Richard Kiplimo, Cheryl Kerama, Khai Hoan Tram, Erick Nyadimo, Martha Njoroge, Dickson Kirathe, David J. Horne.

**Formal analysis:** Jane R. Ong'ang'o, Jennifer Ross, Richard Kiplimo, David J. Horne.

**Funding acquisition:** Thomas R. Hawn, Videlis Nduba, David J. Horne.

**Investigation:** Jane R. Ong'ang'o, Jennifer Ross, Richard Kiplimo, Cheryl Kerama, Khai Hoan Tram, Jerry S. Zifodya, Nellie Mukiri, Erick Nyadimo, Martha Njoroge, Immaculate Kathure, Dickson Kirathe, Thomas R. Hawn, Videlis Nduba, David J. Horne.

**Methodology:** Jane R. Ong'ang'o, Cheryl Kerama, Khai Hoan Tram, Nellie Mukiri, Erick Nyadimo, Martha Njoroge, Immaculate Kathure, Dickson Kirathe, Thomas R. Hawn, Videlis Nduba, David J. Horne.

**Project administration:** Jane R. Ong'ang'o, Cheryl Kerama, David J. Horne.

**Resources:** Thomas R. Hawn, Videlis Nduba, David J. Horne.

**Software:** Dickson Kirathe, David J. Horne.

**Supervision:** Jane R. Ong'ang'o, Cheryl Kerama, Aiban Ronoh, Immaculate Kathure, Dickson Kirathe, Videlis Nduba, David J. Horne.

**Validation:** Jane R. Ong'ang'o, Jennifer Ross, Richard Kiplimo, Cheryl Kerama, Khai Hoan Tram, Jerry S. Zifodya, Martha Njoroge, Aiban Ronoh, Immaculate Kathure, Dickson Kirathe, Videlis Nduba, David J. Horne.

**Visualization:** Jennifer Ross, Richard Kiplimo, Cheryl Kerama, Jerry S. Zifodya, Aiban Ronoh, Immaculate Kathure, Thomas R. Hawn, Videlis Nduba, David J. Horne.

**Writing – original draft:** Jane R. Ong'ang'o, Jennifer Ross, Jerry S. Zifodya, David J. Horne.

**Writing – review & editing:** Jane R. Ong'ang'o, Jennifer Ross, Richard Kiplimo, Cheryl Kerama, Khai Hoan Tram, Jerry S. Zifodya, Nellie Mukiri, Erick Nyadimo, Martha Njoroge, Aiban Ronoh, Immaculate Kathure, Dickson Kirathe, Thomas R. Hawn, Videlis Nduba, David J. Horne.

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
