## [Decision Letter · Decision Letter 0]

28 Oct 2024

PGPH-D-24-02260

Persistently high TB prevalence in Nairobi County neighbourhoods, 2015-2022

Dear Dr. Ong'ang'o,

Thank you for submitting your manuscript to PLOS Global Public Health. After careful consideration, we feel that it has merit but does not fully meet PLOS Global Public Health’s publication criteria as it currently stands. Therefore, we invite you to submit a revised version of the manuscript that addresses the points raised during the review process.

We look forward to receiving your revised manuscript.

Kind regards,

N. Sarita Shah

Academic Editor

Journal Requirements:

Additional Editor Comments (if provided):

Reviewers' comments:

Reviewer's Responses to Questions

**Comments to the Author**

1. Does this manuscript meet PLOS Global Public Health’s publication criteria ? Is the manuscript technically sound, and do the data support the conclusions? The manuscript must describe methodologically and ethically rigorous research with conclusions that are appropriately drawn based on the data presented.

Reviewer #1: Yes

Reviewer #2: Yes

2. Has the statistical analysis been performed appropriately and rigorously?

Reviewer #1: Yes

Reviewer #2: Yes

3. Have the authors made all data underlying the findings in their manuscript fully available (please refer to the Data Availability Statement at the start of the manuscript PDF file)?

Reviewer #1: Yes

Reviewer #2: Yes

4. Is the manuscript presented in an intelligible fashion and written in standard English?

Reviewer #1: Yes

Reviewer #2: Yes

5. Review Comments to the Author

Reviewer #1: STRENGTHS

1. Topic Relevance: The topic of the article is highly relevant, as the persistently high incidence of TB in Nairobi County continues to pose a significant public health challenge in Kenya. Nairobi's urban environment, characterized by high population density and socio-economic disparities, creates conditions that contribute to TB transmission, making the study particularly timely and important. Focusing on nine clusters within the city offers valuable localized insights, allowing for a more targeted analysis of the factors driving TB prevalence in specific areas. By concentrating on these neighborhoods, the article provides a more granular view of the epidemic, which is crucial for designing interventions that address the unique challenges of urban TB control.

2. Clear Presentation of Findings: The article presents its findings in a clear and concise manner, primarily using tables to effectively summarize the data. This structured approach allows readers to easily interpret the key points and trends from the surveys conducted. The use of tables ensures that complex data, such as diagnostic results, prevalence rates, and outcomes, is accessible and straightforward. This clear presentation helps to highlight the main findings without overwhelming the reader with excessive detail, making it an effective format for conveying the study’s results and comparisons between the 2015 and 2022 surveys.

3. Comprehensive Data: The article offers a comprehensive comparison between two surveys, providing valuable data to assess TB trends in Nairobi County neighborhoods over time. The 2022 survey serves as a critical benchmark, allowing programmers to evaluate the effectiveness of TB interventions implemented since the 2015 survey. This longitudinal approach strengthens the analysis by showing whether the strategies used such as expanded diagnostic tools, screening methods, treatment protocols, or community interventions have had any meaningful impact on reducing TB prevalence. By comparing these two data points, the article enables a clear assessment of progress or ongoing challenges. This thorough examination of both surveys not only highlights successes but also identifies areas where interventions may need to be intensified or re-strategized.

4. TB diagnostic tools: The article highlights the effective use of a combination of diagnostic tools, GeneXpert, culture tests, and Chest X-ray in the survey, which enhances the accuracy of TB detection. This multi-faceted approach is advantageous because each diagnostic method has its strengths. A notable improvement in this survey is the further analysis of trace-positive results using culture, which adds depth to diagnostic insights compared to the 2015 survey. The use of these results helps refine diagnostic sensitivity, especially in patients with low bacterial loads. Additionally, the study adapted the symptom screening criteria by broadening it to "any cough" rather than the stricter criterion of cough lasting over two weeks, as was used in the 2015 survey. This change likely increased the sensitivity of case detection, capturing more individuals with early-stage or mild TB symptoms who might otherwise have been missed.

5. Focus on Populations of interest: The article ability to highlight population of interest, particularly the higher concentration of TB cases among men aged 45-54 who are unemployed, is critical. These findings emphasize the need for future targeted interventions, as these demographic appear to be disproportionately affected by TB. This population-specific focus is vital for tailoring public health strategies, as it underscores the importance of addressing socio-economic determinants of health and the role they play in disease transmission. By identifying key groups at higher risk, such as middle-aged men in economically disadvantaged neighborhoods, the article provides actionable insights that can guide more effective and focused TB control measures. These insights will be especially useful for policymakers and health practitioners aiming to design interventions that prioritize the most vulnerable and affected populations.

6. Policy and Programmatic Recommendations: The article provides valuable policy and programmatic recommendations based on its findings, particularly in developing TB case-finding strategies tailored to behavioral patterns and epidemiological variations. This is especially important for addressing TB in specific populations, such as men aged 45-54, who are disproportionately affected. By understanding behavioral patterns such as healthcare-seeking behaviors and the unique epidemiological profile of different groups, the article suggests that TB programs can be more inclusive and effective.

Areas for Improvement and Recommendations

1. Contributing Factors: The article would benefit from a more in-depth analysis of the underlying causes behind the persistently high TB prevalence in the nine clusters studied. It would be valuable to explore factors such as overcrowding, poor ventilation, poverty, and limited access to healthcare in these neighborhoods. This deeper understanding could help identify more targeted interventions and address the root causes driving TB transmission in these high-risk areas.

2. Comparative Analysis with Other Regions: Including a comparative analysis between Nairobi and other urban centers in Kenya could provide broader context. Is TB prevalence also high in other cities, or is Nairobi an outlier? This comparison could help identify whether the issue is unique to Nairobi’s urban dynamics or part of a larger regional trend. Lessons from cities with lower TB rates could offer strategies for mitigating the prevalence in Nairobi’s clusters.

3. Impact of COVID-19 and the Kenyan Health Worker Strike: The article briefly mentions the effects of COVID-19 and the health worker strike on TB detection and treatment. A more detailed exploration of how these events impacted TB case detection, treatment adherence, and overall healthcare access would be beneficial. Such analysis could provide insights into how health system disruptions contribute to TB burden, particularly in urban settings like Nairobi.

4. Data Visualization: To enhance the clarity and impact of the findings, the inclusion of data visualizations, such as heat maps showing TB prevalence across the nine clusters, would be highly useful. These visual tools would allow readers to quickly grasp spatial and temporal trends in TB prevalence and would be particularly impactful when presenting the data at conferences or public health forums.

Reviewer #2: General comments:

- Overall, the paper is well-written. The author clearly outlines the rationale and objectives of the paper, which is to evaluate the prevalence of TB in Nairobi County through the serial surveillance of TB using a population-based survey. The discussion section can be strengthened by stating the local drivers of consistently high TB prevalence in Nairobi, as indicated by the survey. Also, the author may provide examples of evidence-based interventions that have proven effective to reduce the prevalence of TB in similar settings. This is helpful information to help the reader gain a better understanding of the public health implications of the paper’s findings.

Introduction:

1. The author’s hypothesis statement, lines 68-69, is not clearly written. Please revise, as it seems that part of the sentence was cut out.

Methods:

1. In line 97 where it reads ‘The household-level survey procedures were similar for both surveys’, please cite the protocol for the 2015-2016 TB prevalence survey or another publication where the survey procedures methods have been summarized. When referring to methods explained elsewhere, please include a citation so that readers may refer to that work.

2. Can you explain why information on diabetes was not collected in the 2015-2016 survey?

Discussion:

1. In line 330, it is mentioned that ‘local drivers of the TB epidemic are stable and may be used to develop targeted TB control interventions.’ Did the survey highlight what these drivers may be? It is helpful to clearly state what the results show to be the local drivers of TB and provide examples of interventions, supported by the literature, that may successfully reduce TB prevalence in this context.

6. PLOS authors have the option to publish the peer review history of their article (what does this mean? ). If published, this will include your full peer review and any attached files.

**Do you want your identity to be public for this peer review?** For information about this choice, including consent withdrawal, please see our Privacy Policy .

Reviewer #1: **Yes: ** Fadimatu S. Mishara

Reviewer #2: **Yes: ** Aderonke S. Ajiboye

---

## [Editor Report · Decision Letter 1]

7 Jan 2025

Persistently high TB prevalence in Nairobi County neighbourhoods, 2015-2022

PGPH-D-24-02260R1

Dear Dr Ong'ang'o,

We are pleased to inform you that your manuscript 'Persistently high TB prevalence in Nairobi County neighbourhoods, 2015-2022' has been provisionally accepted for publication in PLOS Global Public Health.

Best regards,

N. Sarita Shah

Academic Editor